# Using Small Area Prevalence Survey Methods to Conduct Blood Lead Assessments among Children

**DOI:** 10.3390/ijerph19106151

**Published:** 2022-05-18

**Authors:** Kathryn B. Egan, Timothy Dignam, Mary Jean Brown, Tesfaye Bayleyegn, Curtis Blanton

**Affiliations:** 1Division of Environmental Health Science and Practice, National Center for Environmental Health, Centers for Disease Control and Prevention, Atlanta, GA 30341, USA; bvy7@cdc.gov; 2Office of Community Health and Hazard Assessment, Agency for Toxic Substances and Disease Registry, Centers for Disease Control and Prevention, Atlanta, GA 30341, USA; 3Office of the Director, National Center for Environmental Health, Centers for Disease Control and Prevention, Atlanta, GA 30341, USA; ted9@cdc.gov; 4Department of Social and Behavioral Sciences, Harvard T.H. Chan School of Public Health, Boston, MA 02115, USA; mjb686@outlook.com; 5Division of Global Health Protection, Center for Global Health, Centers for Disease Control and Prevention, Atlanta, GA 30341, USA; cjblantoncdc@gmail.com

**Keywords:** blood lead levels, cluster sampling, children, environmental health, lead exposure, small area prevalence surveys, surveillance

## Abstract

Introduction: Prevalence surveys conducted in geographically small areas such as towns, zip codes, neighborhoods or census tracts are a valuable tool for estimating the extent to which environmental risks contribute to children’s blood lead levels (BLLs). Population-based, cross-sectional small area prevalence surveys assessing BLLs can be used to establish a baseline lead exposure prevalence for a specific geographic region. Materials and Methods: The required statistical methods, biological and environmental sampling, supportive data, and fieldwork considerations necessary for public health organizations to rapidly conduct child blood lead prevalence surveys at low cost using small area, cluster sampling methodology are described. Results: Comprehensive small area prevalence surveys include partner identification, background data collection, review of the assessment area, resource availability determinations, sample size calculations, obtaining the consent of survey participants, survey administration, blood lead analysis, environmental sampling, educational outreach, follow-up and referral, data entry/analysis, and report production. Discussion: Survey results can be used to estimate the geographic distribution of elevated BLLs and to investigate inequitable lead exposures and risk factors of interest. Conclusions: Public health officials who wish to assess child and household-level blood lead data can quickly apply the data collection methodologies using this standardized protocol here to target resources and obtain assistance with these complex procedures. The standardized methods allow for comparisons across geographic areas and over time.

## 1. Introduction

Lead adversely affects most body systems [1,2]. In the U.S., approximately 29 million households contain lead hazards and an estimated 400,000 children ages 1–11 years old have blood lead levels (BLL) above 5 µg/dL, which was the CDC blood lead reference value (BLRV) from 2012 to 2021 [3,4,5]. In 2021, the BLRV for children was lowered to 3.5 µg/dL, based on blood lead data from children age 1–5 years old sampled in the 2015–2018 National Health and Nutrition Examination Survey (NHANES) [6]. Lead has negative effects on cognitive function and attention-related and behavioral problems in children, and these effects may persist into adulthood [7,8]. Even low levels of exposure, including BLLs of <5 and <10 µg/dL, have been associated with academic performance decreases in school-aged children [9]. Maintaining the capacity to respond to children with elevated BLLs, targeting screening to at-risk populations and identifying lead “hotspots” is crucial to prevention efforts. Hotspots are geographic areas with children who have elevated BLLs due to, for example, lead smelting sites, electronic waste recycling or usage of consumer products with lead. State and local health departments can use small area prevalence surveys using cross-sectional and multi-stage cluster survey sampling design to estimate the prevalence of BLLs in a target geographic population such as a town, zip code, neighborhood or census tract. In addition, properly designed and executed small area prevalence surveys are useful and cost efficient tools to identify risk factors and the geographic distribution of BLLs [10]. 

Small area prevalence surveys quickly assess how many individuals are affected by a disease or exposure of interest in a given geographic area. They help guide evidence-based public health decisions. For example, prevalence survey findings of low BLLs may support changing policy from mandatory BLL testing among all children to targeted screening of specific communities [11]. Small area prevalence surveys also supplement or complement local BLL screening data and national surveys such as NHANES. They have detected occupational take home lead exposures [12] and areas with a disproportionate share of elevated BLLs [13]. Research staff for small area prevalence surveys collect household environmental samples and assess potential lead exposures in a specific geographic area [14]. This article documents methods and procedures for community childhood small area prevalence surveys when BLL data are insufficient or absent to calculate the prevalence of elevated BLLs in areas with known point sources of exposure. Thus, small area prevalence survey data are essential to public health response to monitor ongoing or emergent lead exposure. 

This article presents an overview of the requirements for proper planning, data collection, and data analysis and describes the usefulness of small area, point estimate prevalence surveys using two-stage cluster sampling to determine BLLs for children in a specific geographic area. 

## 2. Materials and Methods

Small area prevalence surveys for BLLs are designed to obtain unbiased, representative BLL prevalence estimates, describe the distribution of BLLs and estimate the prevalence of BLLs in a specific geographic area. Specific uses could include determining BLLs at or above the CDC blood lead reference value (≥3.5 µg/dL) [6], and other BLL thresholds such as ≥5 µg/dL or ≥10 µg/dL, and to identify risk factors for lead exposure among children 9–71 months of age (or the age range of interest) in specific geographic areas. Although other survey designs are possible, the two-stage sampling approach using the Community Assessment for Public Health Emergency Response (CASPER) surveys (Table 1) has been widely field tested [15]. The following are the steps to follow:

### 2.1. Preparation

At the start, researchers should define the target population, sample size, estimated response rate, and field work duration [16]. Some surveys can be conducted in 2–3 weeks of fieldwork. Increasing the number of field teams often decreases fieldwork duration.

### 2.2. Survey Area

The survey area is the geographic location of the population of interest. It can be a political designation such as a neighborhood, city, county, community, territory or state or a radius measurement could be used around a point source of interest [17,18,19]. The setting, the specific population impacted and the estimated prevalence of the exposure of interest in that population, and the desired statistical power of the survey should determine the final survey area size. If household is the primary sampling unit (PSU), and there are <800 households, a simple random sample can be used. The random sample is drawn from a complete enumeration of the number of households in the population of interest in the designated area. In general, a cluster sampling strategy is better suited to areas with ≥800 sampling units where enumeration of the population is cost prohibitive. Clusters are small, clearly defined numbers of households within a geographic area. This allows selection of groups of households while still maintaining a probability sample. For the larger samples, clusters are the PSU and selected by probability proportional-to-size sampling for the first stage [20]. Then, within a selected cluster, households are selected, either using systematic or simple random sampling for the second stage.

### 2.3. Survey Population

The survey population is defined by characteristics of interest including child age, length of time in a household or proximity to a point source, parental occupation or hobby using lead, or being a member of an ethnicity with known risk factors for lead exposure. For childhood small area prevalence surveys, children 9-71 months old are selected since BLLs peak during early childhood. If more than one child 9–71 months old lives in the household, the younger child should be selected. A child can be defined as a resident after living in the community for a defined period of time as determined by the researcher (e.g., 6 months). Children who live in multiple households (i.e., are in shared custody arrangements) are eligible if they reside in the selected household for a previously defined minimum period of time (e.g., 2 days/week). Different sampling strata may be necessary for target populations that can be sampled based on other risk characteristics such as age of housing, distance from a lead point source, or other environmental factors. To select children at risk of elevated BLLs, oversample areas with known lead exposure risk factors [15,21]. Survey participation is voluntary, and oversampling may need to occur if the survey response rate is low.

### 2.4. Data Sources

Census data provide the best estimates for the number of young children living in a small area. Children ages 9–71 months old are typically at highest risk of lead exposure and are, therefore, often the population of interest in small area prevalence surveys of childhood blood lead levels. Additional sources for estimating the number of children include immunization records, birth certificate data, elementary school enrollment data and the American Community Survey [22]. In the absence of child-based data, it is necessary to obtain an enumeration of the entire target population. 

### 2.5. Sample Size Calculation

Sample size calculations, carried out prior to conducting the survey, estimate the number of participants required to find a significant association at a chosen α level (e.g., α = 0.05 or 0.1) between BLLs and environmental exposures or to estimate the precision around a targeted estimate [23]. If only a point estimate is needed, a large enough sample is calculated for a targeted margin of error (95% confidence interval) around the geometric mean or the prevalence estimate given as a percentage. The formula below is the simple random sample size calculation for a prevalence point estimate accounting for the population size. This calculation assumes normal distribution.
$$n=\frac{\mathrm{DE}ff\times {\;Z}_{\frac{\alpha }{2}}^{2}\times N\times p\left(1-p\right)}{{\mathrm{ME}}^{2}\times \left(N-1\right)+Z_{\frac{\alpha }{2}}^{2}\times p\left(1-p\right)}$$
where:
Z = value from the standard normal distribution;α = type I error;*N* = population size (for finite population correction factor);*p* = estimated prevalence (hypothesized % of children with BLLs above the CDC blood lead reference value in the survey population);d = 95% confidence limits as percentage of 100 (absolute +/− %);DE*ff* = Design Effect (=1 for random sample);ME = Margin of Error.

The design effect (DE*ff*) for cluster samples is derived from the equation below and represents the increase in variance arising from the cluster design (over the simple random sampling). The sample size is calculated to account for this variance. In this equation, *b* represents the average number of persons to be selected from each cluster and ρ represents the intra-class correlation.
DEff=1+(b−1)ρ

Generally, you will need a larger sample size when the estimated prevalence is very small.

Response rates are the number of participants compared to the total number of people asked to participate. The sample size calculated using the equation above should be increased based on the estimated response rate [24]. At the conclusion of the survey, response rates can be calculated and are important to determine the representativeness of participants. The number of expected households needed is obtained by dividing the estimated number of children aged 9–71 months old living in each household when all children in a household are sampled by the number of households eligible plus the number of households whose eligibility was undetermined. This information is available from U.S. Census or immunization data. 

### 2.6. Cluster Sample 

When random or systematic sampling is not feasible, a cluster survey design can be applied [25,26,27]. The goal of the population-based cluster survey design when studying children’s blood lead levels is to randomly select households with children ages 9–71 months old from the larger survey area. The cluster sample is designed so that every child has an approximately equal probability of selection. The cluster survey design follows the World Health Organization’s Expanded Programme on Immunization (EPI) model, but the accuracy has been improved based on the recommendations of Brogan and colleagues [28].

### 2.7. Sampling Frame for the Cluster

The sampling frame of the cluster includes those units of interest that form the cluster. These units may include households or block groups holding the desired characteristic of interest. During the first stage of a cluster sampling, a list of clusters and their population size are enumerated. Census data are commonly used to identify the sample frame used to define a cluster as well as to provide an estimate of the number of units within that cluster, for example, the number of households within a cluster. Other examples of how census data are used to further refine the sampling frame that is used to define the cluster include stratifying the data by age, gender, and household numbers at the census block or track level. This helps to obtain clusters with the desired units of interest. 

### 2.8. Stages of Cluster Sampling: Stage One

The first stage of cluster sampling involves defining the geographic population of interest and dividing that population into mutually exclusive units to form the cluster sampling frame. From the sample frame, each cluster is defined, enumerated and then selected (Figure 1). Cluster selection is probabilistic and often based on similar, previous surveys of the outcome of interest or on the proportion of individuals in the sub-populations of interest. If estimates from certain sub-populations are desired, then oversample clusters with the sub-populations. This may involve explicitly stratifying clusters with units including the population of interest. Ideally, before any sampling occurs, clusters should be mapped and listed. If the cluster is too large to map, then random segments should be mapped and listed. For rapid surveys, the mapping and listing can be skipped to save time and resources. 

### 2.9. Stages of Cluster Sampling: Stage Two

In the second stage and depending on the necessary sample size, households are systematically selected in each of the clusters [25]. The cluster is the PSU, and the household is the secondary sampling unit. A sample of households from the stage two sampling frame is selected from within each cluster by random selection. Random selection entails selecting each household with equal selection probability. Once the first household has been selected, the next household on the sampling frame is selected below the starting point from which the first was selected (k − 1). This is carried out until the sample size in each cluster is achieved. Once the desired number of households is achieved within each cluster, households are systematically visited until children 9–71 months of age are identified and enrolled. The cluster survey sample size calculation for a prevalence point estimate is shown below:$$n=\frac{\mathrm{DE}ff\times \;t_{df,\frac{\alpha }{2}}^{2}\times N\times p\left(1-p\right)}{{\mathrm{ME}}^{2}\times \left(N-1\right)+t_{df,\frac{\alpha }{2}}^{2}\times p\left(1-p\right)}$$
where:
*t* = *t*-value where degrees of freedom is the number of clusters minus 1 or the number of clusters minus the number of strata;α = type I error;*N* = population size (for finite population correction factor);*p* = estimated prevalence (hypothesized % of children with BLLs above the CDC blood lead reference value in the survey population);d = 95% confidence limits as percentage of 100 (absolute +/− %);DE*ff* = Design Effect (=1 for random sample);ME = Margin of Error.

### 2.10. Minimizing Bias

To reduce bias, households in a cluster should be enumerated to account for any changes in population. Avoid using convenience or sequential sampling.

### 2.11. Fieldwork Considerations

Field staff collect and safeguard survey data (Figure 2). Both the sample size and the travel time to the clusters will determine the number of teams needed. Each team has three or four members: a communications expert, a logistics coordinator, an environmental sampler, and a pediatric phlebotomist. Recruiting field staff from the sample population minimizes cultural and logistical issues and improves participation and response rates. The field team composition, hours of work required, language proficiencies, and travel logistics are important considerations. It is imperative to train field teams to refer residents to important public health services other than lead. Field team safety, such as infectious disease prevention and identifying potentially unsafe field situations, should be considered during planning. Previous studies have partnered with community groups and law enforcement before and during fieldwork [29].

### 2.12. Tracking Documentation

The number of people contacted for participation, the number of people who refused to participate and the number of interviews completed need to be tracked electronically or on paper (Appendix A). For participants, use unique sample identification numbers to ensure that questionnaire data, environmental samples, and blood samples can be matched by participant. Each household, participant, and sample(s) need a linked unique identifier. Barcode labels that include study and participant identification number and the date and time of specimen collection streamline tracking. Determine the minimum number of labels required for each household prior to starting the survey. Field staff will need additional labels in case some are damaged in the field. The ink on the labels may be erased if they are exposed to alcohol wipes.

### 2.13. Community Partners and Participant Compensation

Relevant stakeholders (state/local health departments, neighborhood groups, non-governmental organizations, colleges/universities, and federal partners) should be engaged during the early stages of survey preparation. Community awareness by sharing a one-page fact sheet with the community and potential study participants will increase survey success. Carefully consider the benefits of some form of financial compensation for participants (e.g., gift cards). This involves balancing respect for participants’ time with the potential for undue influence to coerce participation.

### 2.14. Institutional Review Board (IRB) Documentation

Consult with the appropriate IRB before study initiation. IRB review and approval may be necessary because of real or perceived risks of participation (e.g., venous blood is often collected). Individual IRBs at the affiliated university or public health institution can determine if a survey qualifies for an exemption or an expedited review because of minimal risk to human subjects or inability to generalize findings to other communities. 

### 2.15. Consent Forms

Only parents or legal guardians can give consent for participation. Verbal permission needs to be granted for all participants prior to asking eligibility questions. Parents or guardians who provide verbal consent for participation will need to complete and sign a written consent form. Once consent is completed, identify a private place to conduct the interview. Administer interviews exactly as written. If no one is home when eligibility is being determined, leave a resident notecard, mark the location for follow-up, and return in the evening of the same or next day. After two visits with no one home, it will be necessary to enroll a different household. The first house on the left side of the house should be approached; if there is no eligible child at that address, approach the house on the right. Repeat this process until an eligible household is identified.

### 2.16. Data Collection Methods

Data are collected using paper forms, laptops, smartphones or tablets. Paper forms require little training and are inexpensive. However, manual data entry may result in transcription errors. To confirm accuracy of data entry, perform double data entry on 10–15% of the paper records [29]. Electronic devices have higher up-front cost for hardware and software but can streamline interviews, allow for automatic data transfer and GPS locations, and reduce errors. If electronic devices are used, paper forms should be available in case of electronic malfunctions.

### 2.17. Biologic Specimen Collection/Laboratory Considerations

Before commencing the study, determine whether to collect capillary or venous blood samples. Capillary samples can be obtained quickly and easily, and results can be analyzed within a few minutes by portable point-of-care blood lead analyzers and reported immediately (Magellan Diagnostics; Billerica, Massachusetts). Capillary samples have a higher probability of contamination that may result in a positive bias. Venous samples require trained phlebotomists and are considered more invasive than capillary samples. They need to be sent to laboratories for analysis. However, bench laboratory methods cover a wider range of BLLs, have lower limits of detection, are less likely to be contaminated, and have higher quality control standards as they must meet Clinical Laboratory Improvement Amendments (CLIA) certification guidelines. Thus, venous samples are preferred. 

### 2.18. Environmental Sample Collection

Environmental samples include interior dust, exterior soil and water. Home location is recorded using a geographic positioning system (GPS) reading at the front door at the time of sampling. Electronic devices for data collection are useful in that they can record GPS locations automatically. A schematic of the home should be drawn indicating the sampling locations. Samples should be collected concurrently with children’s blood samples. A water sample should be collected from the kitchen or bathroom sink. All researchers should wear powder-free gloves during the collection and water should not be run prior to collecting the sample. Parents and guardians should help to identify the appropriate areas to test (e.gs. child’s play area, and bedroom, windowsills, behind front door, etc.) using dust wipes and composite soil sampling. Onsite paint analyses, use portable X-ray fluorescence analyzers according to the U.S. Department of Housing and Urban Development protocol [30]. These samples can be collected in all households or a random selection of enrolled households, depending on available resources. Survey planners should adapt the sampling strategy to the specific objectives of the study. When all samples have been collected and labeled, they should be transported and included with a chain of custody form sent to the laboratory. 

## 3. Results

### 3.1. Results Reporting

BLL results should be promptly reported to parents/guardians and health care providers. All individualized test results are private and cannot be shared with anyone other than parents or health care providers. BLL results should be explained in person within 72 h of blood draw to each parent/guardian. This timeline may be difficult to adhere to depending on the distance from the sample draw to the regional reference laboratory and laboratory methods needed such as Graphite Furnace Atomic Absorption Spectroscopy (GFAAS) or Inductively Coupled Plasma Mass Spectrometry (ICP-MS) that may not be available at smaller clinical laboratories; thus, adaptations to the timeline can be made situationally. Provide a paper copy of the follow-up results. Environmental sample results take longer to analyze but should be reported to the parents/guardians when available. BLLs above the CDC blood lead reference value should be reported to the child’s parents/guardians and to their health care provider (such as their pediatrician) and the local or state health department as quickly as possible after parent/guardian notification. Elevated BLLs need to be addressed by a physician and blood lead levels over 60 µg/dL are considered an emergency [31]. Provide these to the healthcare provider immediately. Include a recommended follow-up blood lead schedule and conduct an immediate environmental investigation [32].

### 3.2. Data Analyses

Data analysis is more complex for cluster sampling than for simple random sampling. Survey response rates should be calculated to determine the representativeness (bias) of the sample. Use the field tracking form to provide data such as survey completion, contact and cooperation rates. Refusal rates are calculated based on the number of households who refused to join the study divided by the number of households who participated. Information such as accessibility to the selected household, interview completed or not, and refusals should be recorded on the tracking form. If response rates are low or non-response is higher in specific areas/populations, the data may not be representative of the population of interest and external validity may be compromised [33]. Weighting non-response adjustments can be used to attempt to reduce bias. 

For multi-stage samples, non-response occurs at each level of sampling. Thus, response rates need to be calculated at each stage of sampling. Response rate calculations are as follows:$$\mathrm{Cluster}\;\left(\mathrm{PSU}\right)\mathrm{Reponse}\;\mathrm{Rate}\;=\frac{\mathrm{Number}\;\mathrm{of}\;\mathrm{Clusters}\;\mathrm{Responding}}{\mathrm{Total}\;\mathrm{Number}\;\mathrm{of}\;\mathrm{Clusters}\;\mathrm{Selected}}$$
$$\mathrm{Household}\;\mathrm{Reponse}\;\mathrm{Rate}\;=\frac{\mathrm{Number}\;\mathrm{of}\;\mathrm{Households}\;\mathrm{Responding}}{\mathrm{Total}\;\mathrm{Number}\;\mathrm{of}\;\mathrm{Households}\;\mathrm{Selected}}$$
$$\mathrm{Person}\;\mathrm{Reponse}\;\mathrm{Rate}\;=\frac{\mathrm{Number}\;\mathrm{of}\;\mathrm{Persons}\;\mathrm{Responding}}{\mathrm{Total}\;\mathrm{Number}\;\mathrm{of}\;\mathrm{Persons}\;\mathrm{Selected}}$$
$$\begin{array}{ll} \mathrm{Total}\;\mathrm{Reponse} & \mathrm{Rate}\;\\ & =\;\mathrm{Cluster}\;\mathrm{Response}\;\mathrm{Rate}\;\times \;\mathrm{Household}\;\mathrm{Response}\;\mathrm{Rate}\;\\ & \times \;\mathrm{Person}\;\mathrm{Response}\;\mathrm{Rate} \end{array}$$

Sampling weights for both households and children are needed to calculate prevalence estimates and make inferences about the entire population of children 9–71 months of age. Sampling weights can be adjusted to account for unequal selection probability that may have occurred due to changes in the selected clusters’ population, oversampling, and adjustments for non-response and post-stratification. Post-stratification using known population estimates can reduce survey bias. Complex survey procedures in SAS/SUDAAN, STATA, R software (SAS Institute, Inc., Cary, NC, USA; RTI International, Research Triangle Park, NC, USA; R Foundation for Statistical Computing, Vienna, Austria) or EpiInfo software [34] should be used to account for unequal weighting, stratification and clustering in the sample. 

Present household and child demographics and other characteristics using descriptive statistics. Hierarchical multilevel linear regression, logistic regression and 95% confidence intervals are used to determine risk factors that predict elevated BLLs. As observations within a cluster are not independent, estimate the degree of similarity using the intra-cluster correlation coefficient (ICC) and account for correlation when computing the variance either using Taylor Series estimation or replicate weights [35]. Failing to account for the clustering and weighting in modeling can result in incorrect p-values, biased estimates and effect sizes and confidence intervals that are too narrow [36]. Several studies have published guidelines for interpreting ICC [37] and for accounting for clustered data during the analysis [38,39]. Additionally, BLLs are typically right-skewed and should be log-transformed prior to regression analyses. For analyses of geometric means, log-transformed estimates need to be back-transformed prior to interpretation. Bivariate analyses can be used to assess individual risk factors associated with BLLs. Then, multivariate analyses that include confounding factors and interaction terms can be used to predict the adjusted association between BLLs and risk factors of interest. Collinearity between variables in the predictive model can be assessed by variance inflation factors [40].

### 3.3. Other Sources of Data

It may be useful to compare study findings with existing local child blood lead levels if appropriate. However, before starting any such comparisons, it is important that researchers have a good understanding of any previous blood lead testing carried out in the area as direct comparisons across studies where data were collected using different methods are problematic. One example of a potential source for comparison is BLL surveillance data. Survey results can be compared to existing BLL data such as country, state, city or county level children’s BLL surveillance data (or surveillance data based on another geographic boundary). These comparisons can be inaccurate though as existing surveillance data are unlikely to be random or have high screening rates. A key limitation of surveillance data is that program- or clinical office-based data are not population-based. Surveillance data are restricted to individuals targeted for BLL testing by health care providers. Other sources of data to consider include local program case management data of children with elevated BLLs, home inspection data, previous studies conducted in the area, Adult Blood Lead and Exposure Surveillance Data (if occupational “take home” lead exposure is suspected) [41], the Social Vulnerability Index [42] and data from nationally representative studies such as NHANES. NHANES estimates the number of individuals with elevated BLLs. Because small area prevalence survey data are also population-based, they can be compared to NHANES [43], though NHAHES data are not available on a local level.

### 3.4. Information Dissemination

Dissemination of findings is vital following the conclusion of field work. Presentation and interpretation of results help partners to quantify and describe risk factors for child lead exposure. Preliminary findings should be shared with leaders and key stakeholders such as surveyed families within 2–3 days of the completion of field data collection; typically, this is completed via a slide presentation that includes a discussion of key findings, implementation of any recommendations, next steps, and lessons learned. A final report consisting of background, objectives, sampling frame, methods, two-stage cluster design, the questionnaire, the number and training of interview teams, data analysis procedures, a map of the assessment area, initial results (including response rates), key findings in table format, a discussion concerning the main findings, limitations, action-oriented recommendations based on initial results, and acknowledgements of partners should be published. The final report to stakeholders and families can be issued within a couple of months following data collection. Results should also be distributed to clinical health care providers and others with an interest in child health and development in the community.

## 4. Discussion

Small area prevalence studies can identify risk factors and quantify the extent of lead exposure. Cluster sampling surveys carried out in a scientifically rigorous manner with adequate sample size have reasonable precision (±10%) [44]. The advantages of conducting small area prevalence surveys in comparison to other sampling methods include minimal cost and the short amount of time required to complete the survey. Providing training to the field team in developing the survey tool, sampling, interviewing, conducting the survey and analyzing builds local capacity. Results can be used to target lead poisoning prevention and other public health interventions. Small area prevalence surveys are conducted to determine the contribution of environmental lead sources in the population and establish a baseline before preventive action so that impact can be measured, and the efficacy of interventions can be demonstrated.

Results from blood lead small area prevalence surveys fill an important niche in the child BLL surveillance landscape. Point sources of lead, although not significant at the national or state level, can affect communities and residents. Differential lead exposures due to community exposures such as lead-based paint and water may also be missed by national surveys. Recent U.S. surveys (e.g., NHANES) provide an excellent estimate to track national progress towards reducing lead exposure [45]. Small area prevalence surveys can be compared to NHANES data and measured against national benchmarks. State-based surveillance data are not designed for this purpose and are difficult to compare across states or counties because data collection and laboratory methods vary, and health care providers target high-risk children based on location-specific requirements.

Results from small area prevalence surveys often identify not previously known information about the community. A survey conducted in 2010 in Puerto Rico investigated reports of BLLs above 5 ug/dL among Puerto Rican children who were tested while visiting the U.S. mainland. The survey did not identify that being from Puerto Rico was a risk factor. However, the study identified that local battery recycling employees were bringing lead home on their clothes and exposing children at home to lead [12]. Similarly, a small area prevalence survey in 2001 identified two Chicago communities with twice the prevalence of children with BLLs above the CDC blood lead reference value at the time compared to Chicago’s citywide surveillance data [13]. The study also discovered that parent-reported immunization levels were much lower than expected [46]. Finally, a 2014 blood lead survey conducted in Philadelphia led to several subsequent community-requested, educational Soil Screening, Health, Outreach and Partnership (soilSHOPs) events [14,47].

The prevalence study methods outlined here are subject to limitations. It may be difficult to meet the sample size recommendations and enroll the necessary number of participants. In addition, there may be selection bias among those who self-select to enroll, which limits the representativeness of the survey. Data collection may be incomplete if parents/guardians’ consent to completing the survey and environmental samples but not blood sampling. Enroll more children than the predetermined required sample size addresses this. To assess possible differences between families who do and do not participate in studies, demographic information about non-respondents can be collected and compared with participating families.

## 5. Conclusions

Small area prevalence surveys are a valuable tool in identifying inequitable lead exposures. They can help states and local programs to develop the capacity to respond to children with elevated BLLs, target screening to at-risk sub-populations, and identify lead “inequitable hotspots” and emerging sources or high-risk populations that need primary and secondary prevention efforts. Public health practitioners who wish to assess child and household-level environmental and BLL data can quickly apply the data collection methodology described. Maintaining capacity to respond to elevated BLLs, targeting screening to at-risk populations, and identifying lead “hotspots” are crucial to primary and secondary prevention efforts. Considering recent research demonstrating that there is no known safe BLL threshold for children, prevalence surveys are a useful adjunct to surveillance data for areas or populations that may not be well-represented in surveillance data.

## Figures and Tables

**Figure 1 ijerph-19-06151-f001:**
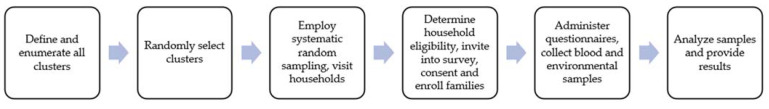
Cluster sampling flow chart: stage one and stage two.

**Figure 2 ijerph-19-06151-f002:**
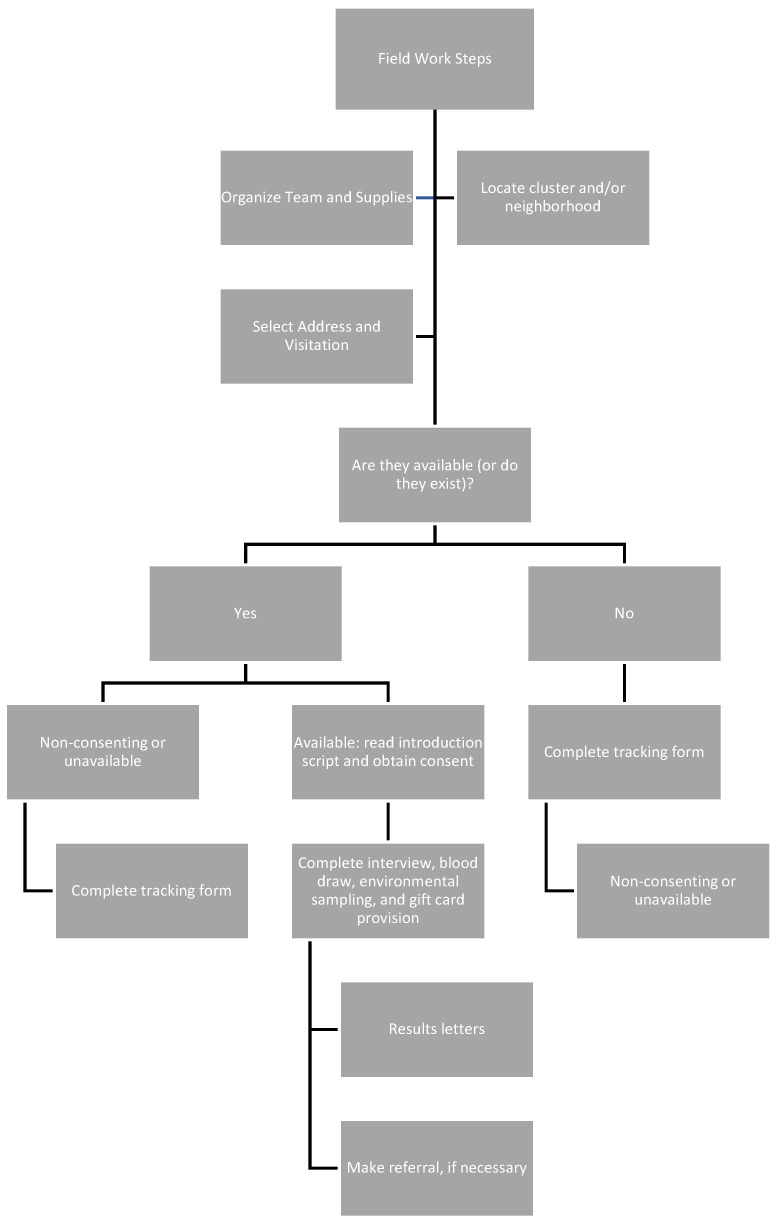
Flow chart of field work steps.

**Table 1 ijerph-19-06151-t001:** Summary of Community Assessment for Public Health Emergency Response (CASPER) procedures.

**Phase I: Prepare for CASPER**	
Define Objectives	- Define the study objectives
Sampling and Mapping	- Define the geographic area - Secure required data for cluster sampling - Determine clusters’ boundary (i.e., block group or block) and sampling unit for cluster selection - Randomly select 30 clusters - Generate cluster maps by using the Census website, GIS software or GPS
Materials and logistics	- Develop the data collection instrument and database - Develop data entry platform - Prepare the tracking form - Prepare the referral form - Prepare the consent form - Identify field coordination center - Organize the assessment teams - Provide training for field teams - Prepare supplies and other assessment materials
**Phase II: Conduct the Assessment**	
Cluster	- Navigate to the cluster - Randomly choose the starting point - Count the houses in the cluster and divide by 7 to determine the sample interval - Select systematically the 7 households and select the individuals to interview
Interviews	- Introduce the team and read the verbal consent - Conduct the interview - Complete the tracking form - If necessary, complete the referral form and send it to the designated person - Hand out public health materials
**Phase III: Analyze Data**	
Data Management	- Enter data - Merge and clean entered data
Data Analysis	- Generate unweighted and weighted frequencies, percentages, and confidence intervals - Interpret the findings
**Phase IV: Write the Report**	
Write	- Write preliminary report within 72 h of the assessment during disaster situations - Write the final report
Report Dissemination	- Disseminate the report findings to stakeholders

## Data Availability

Not applicable.

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
