# Peer review of "Using Small Area Prevalence Survey Methods to Conduct Blood Lead Assessments among Children"

_ijerph, 2022, doi:10.3390/ijerph19106151_

Round 1

Reviewer 1 Report

Major concern

The lack of clear experimental methods and results may make it unsuitable for an original article. The authors may try to rewrite it as a review article.

Author Response

Thank you for your review of the article. The article is original research in that it is publishing new guidance on methods for how to conduct a small area prevalence study of blood lead levels among children. This is not a review of existing literature or studies. However, we do cite previous articles that employ the prevalence study methods described in our article. The citations include references such as 13, 14, 29 & 46.

Reviewer 2 Report

The manuscript of Egan et al presents a study on using small area prevalence survey methods to conduct blood lead assessments among children. Some remarks are as follows:

1. The Abstract: some words are missing in line 37. Surely the sentence is not supposed to end in "Public Health Implications". The last few lines of the Abstract (lines 38-41) have to be re-formatted (in terms of font size) and connected to the sentences that go before.

2. Figure 1: this should not be a figure. The authors could have simply provided its contents as two formulas in the manuscript.

3. Figure 2: re-formatting is necessary, the current version of this figure looks as if it were not well prepared by the authors.

4. Figure 3: the authors could have provided a better-looking and better-prepared version of this.

5. Figure 4: this should not be a figure either. The authors could have provided all its contents as formulas.

6. There is no Conclusion section in this manuscript?

7. The authors should add more recent publications as references for the manuscript. Most citations date back to before 2010, some even in the 1980s, 1990s. The Introduction section should be updated accordingly.

Author Response

Thank you for your review. A point by point response is included below.

The manuscript of Egan et al presents a study on using small area prevalence survey methods to conduct blood lead assessments among children. Some remarks are as follows:

  1. The Abstract: some words are missing in line 37. Surely the sentence is not supposed to end in "Public Health Implications". The last few lines of the Abstract (lines 38-41) have to be re-formatted (in terms of font size) and connected to the sentences that go before.
  • Thank you for your review. This has been updated. “Public Health Implications” is a section within the abstract. The other 3 lines were adjusted to size 9 font to match the rest of the abstract. These resulted from a formatting error when the article was updated from a double-spaced word document to the journal format word document.
  1. Figure 1: this should not be a figure. The authors could have simply provided its contents as two formulas in the manuscript.
  • Figure 1 has been deleted and the formulas have been incorporated into the content of the manuscript.
  1. Figure 2: re-formatting is necessary, the current version of this figure looks as if it were not well prepared by the authors.
  • Figure 2 has been reformatted. It is now called figure 1 as the original figure 1 was incorporated into the text.
  1. Figure 3: the authors could have provided a better-looking and better-prepared version of this.
  • This is now Figure 2. We renamed the figure to indicate that it is a simple flow chart. The submitted manuscript text file included the figures at the end of the file rather than in text. The figures were added within the manuscript text after the manuscript was formatted for journal specifications. All figure formatting reviewer comments have been addressed and did not exist prior to the files being combined.
  1. Figure 4: this should not be a figure either. The authors could have provided all its contents as formulas.
  • The original Figure 4 has been deleted and the formulas have been incorporated into the content of the manuscript.
  1. There is no Conclusion section in this manuscript?
  • The conclusion is the public health implications section. We have added “Conclusions” to the title of that section.
  1. The authors should add more recent publications as references for the manuscript. Most citations date back to before 2010, some even in the 1980s, 1990s. The Introduction section should be updated accordingly.
  • Our manuscript reviews the historical use of prevalence studies (and thus citing older studies) to set the foundation for the contemporary methods that we describe in the manuscript to assess lead exposure. There are many up to date publications cited such as those published in the last 5 years.

Reviewer 3 Report

the paper is well written with a clear focus on the research framework. the introduction is informative and gives enough information about the research. the methodology is well described with enough information for anyone who wants to repeat a similar study. the results are well described and satisfactory disscused.

Reading the paper concluded that the framework of the study was to find screening methods for early response for the increasing the lead concentration in children’s blood in specific area. Generally the authors should be briefly  explain  the importance of early detection of increased lead levels in children’s blood and how they influence on their development.

 When I evaluate the novelty of the paper I   compile the strict rules that any information that could led to new findings or improve already known conclusion as look as original and hence this paper have potential to improve selected research topic.

The paper is focused on small area hence it could be regarding as case study or pilot research for the specific sites or to see how accident influence the population on specific sites.

The first of all the authors should be more precise in explanation how they determined the specific small area use in investigation i.e. what was the criteria for selecting the small area. In the line 137 the authors gave formulas for sampling size and cluster sample I have questions: “Why the authors assumed that samples apply normal distribution?” it’s necessary to confirm that assumption with appropriate statistical test. In addition the paper will be improved compare test group with control group. If they cannot for, control group comparison with referent value could be also used.

As I understand the study is based on voluntary participation, and they didn’t cover risk if someone from the selected research area doesn’t want to participate in the study.

Also I have concern about number of samples in text the author said the number of household samples was 7 isn’t it too small for adequate conclusion? Please explain.

The table 1 is redundant and hard to follow hence I suggest authors to divide it in several tables i.e. every phase of the protocol should be separate table.

The figure 1 give formula in my opinion is much better to give it as equation similar suggestion is for figure 4 also. Figure 3 describes scheme of sampling well and I don’t have any concern about it.  

in my opinion, this study could be the starting point for screening methods for early detection of increase of lead blood level in childrens.

Author Response

Thank you for your review. A point by point response is below.

the paper is well written with a clear focus on the research framework. the introduction is informative and gives enough information about the research. the methodology is well described with enough information for anyone who wants to repeat a similar study. the results are well described and satisfactory disscused.

Reading the paper concluded that the framework of the study was to find screening methods for early response for the increasing the lead concentration in children’s blood in specific area. Generally the authors should be briefly  explain  the importance of early detection of increased lead levels in children’s blood and how they influence on their development.

  • We have added two sentences in the introduction about the health effects of lead exposure: “Lead has negative effects on cognitive function and attention-related and behavioral problems in children and these effects may persist into adulthood [7,8]. Even low levels of exposure, including BLLs of <5 and <10 µg/dL, have been associated with academic performance decreases in school-aged children [9].” (lines 51-55)

 When I evaluate the novelty of the paper I   compile the strict rules that any information that could led to new findings or improve already known conclusion as look as original and hence this paper have potential to improve selected research topic.

The paper is focused on small area hence it could be regarding as case study or pilot research for the specific sites or to see how accident influence the population on specific sites.

The first of all the authors should be more precise in explanation how they determined the specific small area use in investigation i.e. what was the criteria for selecting the small area.

  • The survey area is discussed in detail from lines 107-113. “The survey area is the geographic location of the population of interest. It can be a political designation such as a neighborhood, city, county, community, territory or state or a radius measurement could be used around a point source of interest. The setting, the specific population impacted and the estimated prevalence of the exposure of interest in that population, and the desired statistical power of the survey should determine the final survey area size.”

In the line 137 the authors gave formulas for sampling size and cluster sample I have questions: “Why the authors assumed that samples apply normal distribution?” it’s necessary to confirm that assumption with appropriate statistical test.

  • A normal distribution was assumed as this is the formula for a simple random sample size calculation. Simple random samples assume normal distributions. We added clarification of this on line 155, ‘This calculation assumes normal distribution.’

In addition the paper will be improved compare test group with control group. If they cannot for, control group comparison with referent value could be also used.

  • The use of a control group is discussed in “other sources of data” on page 12, line 402 and is often not appropriate for small area prevalence studies of children’s BLLs. Existing surveillance data is unlikely to be random, population based or even of a moderately high rate. Thus comparisons would not be valid. It is important that researchers have a good understanding of any previous blood lead testing done in the area. However, direct comparisons across studies where data were collected using different methods are problematic.  
  • We added “levels if appropriate. However, before starting any such comparisons, it is important that researchers have a good understanding of any previous blood lead testing done in the area as direct comparisons across studies where data were collected using different methods are problematic. One example of a potential source for comparison is BLL surveillance data. Survey results can be compared to existing BLL data such as country, state, city or county level children’s BLL surveillance data (or surveillance data based on another geographic boundary). These comparisons can be inaccurate though as existing surveillance data is unlikely to be random or have a high screening rate.” (lines 403-411) to the text to give examples of appropriate comparison groups when reporting results.
  • We also added “and data from nationally-representative studies such as the National Health and Nutrition Examination Survey (NHANES).” (line 417-419) to clarify the use of nationally-based estimates.

As I understand the study is based on voluntary participation, and they didn’t cover risk if someone from the selected research area doesn’t want to participate in the study.

  • Participation is voluntary. There is no risk if someone does not want to participate in the study. If the response rate is low, the research team will need to sample more households in order to reach the desired sample size. If this is not possible, then sample size limitations need to be addressed in the final report. We added “Survey participation is voluntary, and oversampling may need to occur if the survey response rate is low.” To lines 136-138.

Also I have concern about number of samples in text the author said the number of household samples was 7 isn’t it too small for adequate conclusion? Please explain.

  • The number “7” is for the CASPER summary procedures presented in Table 1 only. These are emergency response procedures and are an existing methodology that we are building on in this manuscript. Their guidance is an example of where to start. We do not suggest a total of 7 household samples for childhood blood lead surveys in this methodology paper. We present the sample size calculations to select an appropriate cluster size based on the desired outcomes.

The table 1 is redundant and hard to follow hence I suggest authors to divide it in several tables i.e. every phase of the protocol should be separate table.

  • We have reformatted Table 1 for clarity. This table is presenting existing procedures for CASPER community assessment summaries which are for disaster response and is not specific to blood lead assessments.

The figure 1 give formula in my opinion is much better to give it as equation similar suggestion is for figure 4 also. Figure 3 describes scheme of sampling well and I don’t have any concern about it.  

  • From your suggestion and another reviewers, we have incorporated the information from the original Figures 1 and 4 into the body of text and deleted them as standalone Figures.

in my opinion, this study could be the starting point for screening methods for early detection of increase of lead blood level in childrens.

Thank you.

Round 2

Reviewer 1 Report

Major concern

In accordance with the instructions given to authors, the sections of a research manuscript should include an introduction, materials and methods, results, discussion, and conclusions.

Author Response

Thank you for your review. We have added the section headers to the abstract and body of the text as requested.